# Racial Differences in Blood Pressure and Autonomic Recovery Following Acute Supramaximal Exercise in Women

**DOI:** 10.3390/ijerph20095615

**Published:** 2023-04-23

**Authors:** Nicole Bajdek, Noelle Merchant, Sarah M. Camhi, Huimin Yan

**Affiliations:** 1Exercise and Health Sciences Department, Manning College of Nursing and Health Sciences, University of Massachusetts Boston, Boston, MA 02184, USA; 2Kinesiology Department, University of San Francisco, San Francisco, CA 94117, USA

**Keywords:** supramaximal exercise, racial difference, blood pressure, autonomic recovery

## Abstract

Despite the growing popularity of high-intensity anaerobic exercise, little is known about the acute effects of this form of exercise on cardiovascular hemodynamics or autonomic modulation, which might provide insight into the individual assessment of responses to training load. The purpose of this study was to compare blood pressure and autonomic recovery following repeated bouts of acute supramaximal exercise in Black and White women. A convenience sample of twelve White and eight Black young, healthy women were recruited for this study and completed two consecutive bouts of supramaximal exercise on the cycle ergometer with 30 min of recovery in between. Brachial and central aortic blood pressures were assessed by tonometry (SphygmoCor Xcel) at rest and 15-min and 30-min following each exercise bout. Central aortic blood pressure was estimated using brachial pressure waveforms and customized software. Autonomic modulation was measured in a subset of ten participants by heart-rate variability and baroreflex sensitivity. Brachial mean arterial pressure and diastolic blood pressure were significantly higher in Blacks compared to Whites across time (race effect, *p* = 0.043 and *p* = 0.049, respectively). Very-low-frequency and low-frequency bands of heart rate variability, which are associated with sympathovagal balance and vasomotor tone, were 22.5% and 24.9% lower, respectively, in Blacks compared to Whites (race effect, *p* = 0.045 and *p* = 0.006, respectively). In conclusion, the preliminary findings of racial differences in blood pressure and autonomic recovery following supramaximal exercise warrant further investigations of tailored exercise prescriptions for Blacks and Whites.

## 1. Introduction

High-intensity interval exercise has demonstrated health benefits including but not limited to improved resting blood pressure, increased aerobic capacity and muscular endurance, enhanced glycemic control, and improved autonomic regulation in healthy and clinical populations [1,2] and may assist in disease or symptom management in those with chronic conditions [1,2,3]. It was suggested that the strategy of alternating between anaerobic exercise and recovery allows individuals to engage in vigorous-intensity exercise, producing a greater hemodynamic strain and metabolic stimulus for cardiovascular adaptations [4]. Additionally, with the growing popularity of high-intensity interval exercise, some individuals may prefer this time-efficient form of exercise over traditional aerobic training programs. However, existing physical activity guidelines only include recommendations for moderate- and vigorous-intensity aerobic physical activity [5]. Anaerobic exercise refers to short periods of intense physical activity fueled primarily by anaerobic glycolysis and phosphocreatine [6]. The inclusion of anaerobic exercise recommendations in physical activity guidelines may assist in disseminating the known benefits of and engagement in this type of exercise. However, little is known about the acute effects of supramaximal exercise on blood pressure or autonomic modulation, which might provide insight into the individual assessment of responses to training load.

Hypertension and cardiovascular disease (CVD) are more severe and prevalent in Black individuals compared to Whites, with similar [7,8,9] or increased severity [9] in women. The age-adjusted prevalence of hypertension is significantly higher among non-Hispanic black women compared to their non-Hispanic White and Hispanic peers [10]. Cardiovascular hyper-reactivity to stress is a major risk factor for hypertension. Even in healthy individuals, Blacks tend to exhibit heightened central aortic blood pressure, lower Nitric Oxide bioavailability and cardiovascular hyper-reactivity to stress compared to their White counterparts, which may contribute to the increased risk of hypertension and cardiovascular abnormalities [11,12].

Following acute aerobic exercise, transient blood pressure reductions of 5–7 mmHg have been observed, which may demonstrate clinical significance as reduced risk for cardiovascular events and assist in predicting chronic blood pressure reductions following exercise training [13,14]. While this acute blood pressure reduction has been consistently observed in Whites following acute aerobic exercise and associated with histamine receptors mediated post-exercise vasodilation [15], it appears to be absent in Black individuals [12]. The lack of post-exercise hypotension in Black individuals could be a result of increased arterial stiffness due to sympathetic activation or compromised sympathetic withdrawal following aerobic exercise and may be partly mediated by histamine receptors [16,17]. Despite the growing popularity of high-intensity interval exercise as a worldwide fitness trend and the inclusion of vigorous-intensity aerobic physical activity in national guidelines [18], few studies have examined the acute effects of supramaximal exercise on recovery blood pressure [19,20,21], and no study has examined the potential racial differences in hemodynamic responses following supramaximal exercise. Therefore, a better understanding of acute changes in blood pressure following supramaximal exercise would be important to outline recommendations for chronic training and provide foundations for individualized exercise prescriptions for Blacks and Whites.

The sympathetic and parasympathetic branches of the autonomic nervous system play key roles in blood pressure and heart rate regulation at rest and during exercise [22,23,24]. Compared to Whites, normotensive Blacks tend to exhibit impaired heart rate variability (HRV) [25] and baroreflex sensitivity (BRS) at rest [22,25,26,27] and with simulated hypo- and hypertension [28] as well as sympathetic responsiveness via exaggerated sympathetic vascular transduction [26], which may contribute to the development of sustained hypertension in normotensive Black women [26]. Acute anaerobic exercise may serve as a physiological stressor and sympathetic stimulus, and the recovery pattern of cardiac autonomic control from acute exercise may provide useful information for evaluating the risks and benefits of this exercise modality. However, to our knowledge, no studies have compared post-exercise autonomic recovery responses between Black and White women. The steep ramp test (SRT) is a ramp exercise test conducted on a cycle ergometer using an increasing workload until the participant achieves volitional exhaustion [29]. It has been used in various populations and serves as a good model to study anaerobic exercise stimulus [29,30,31,32], and the peak power output from SRT is strongly correlated with maximal aerobic power from standard VO_2peak_ tests in clinical populations [31,33].

Therefore, the purpose of this study was to evaluate blood pressure and autonomic recovery responses to acute supramaximal exercise in young, healthy Black and White women, which might provide insight into the individual assessment of responses to training loads. We hypothesized that Black women would not exhibit reduced brachial and aortic blood pressure following repeated bouts of acute supramaximal exercise, while White women would exhibit post-exercise brachial and aortic blood pressure reductions. In addition, Black women would exhibit lower sympathovagal balance and impaired HRV and BRS responses following this type of exercise compared to their White counterparts.

## 2. Materials and Methods

### 2.1. Participants

This study was conducted according to the guidelines of the Declaration of Helsinki and approved by the Institutional Review Board of the University of Massachusetts, Boston (protocol code 2018062 on 29 March 2018). Informed consent was obtained from all subjects involved in the study. A convenience sample of participants was recruited within the University of Massachusetts, Boston and from the greater Boston area through flyers, mass emails to the University of Massachusetts, Boston campus community, and word of mouth.

Twenty females without hypertension (blood pressure < 130/80 mmHg) [34] aged 18–27 years and who self-identified as Black or White participated in this study. All participants were free of cardiovascular, metabolic, renal, or respiratory diseases, and all were non-smokers. Participants taking any medications other than combined estrogen and/or progestin hormonal contraceptive therapy were excluded from participation. Data was collected between 2018 and 2020. Participant recruitment and testing were prohibited between 2020–2022 due to COVID-19-related campus closure and restrictions. Due to emerging research suggesting an impact of COVID-19 on autonomic function, no additional recruitment effort has been made to avoid potential confounding effects from COVID-19.

### 2.2. Study Design

A schematic of the testing protocol is presented in Figure 1. All resting and post-exercise measurements were obtained in the supine position in a quiet, dimly lit room and participants were asked to fast for 3 h, avoid exercise for at least 12 h, and refrain from caffeine and alcohol consumption prior to testing. Following 10 min of rest in the supine position, brachial and aortic blood pressures were measured at rest, and 15 (P15) and 30 (P30) minutes following the first (SRT 1) and the second (SRT 2) bout of supramaximal exercise using the SRT protocol. Autonomic data were obtained at rest and 20 (P20) minutes following each bout of exercise for a subset of participants (Black: *n* = 4, White: *n* = 6). Standardized procedures and protocols were used to ensure consistency across participants and to minimize any potential bias.

### 2.3. Anthropometrics

Height and weight were recorded using a stadiometer and beam balance platform scale (876 Flat scale, Seca, Chino, CA, USA) to the nearest 10th of a centimeter and kilogram, respectively, and body mass index (BMI) was calculated by dividing weight over height squared (kg/m^2^). Moderate to vigorous physical activity (MVPA) was obtained through self-report based on participant responses on their exercise frequency (times/week), duration (minutes), and intensity (moderate or vigorous) to determine total volume per week (minutes/week).

### 2.4. Blood Pressure Measurement

Brachial systolic (SBP) and diastolic (DBP) blood pressures were obtained using an automated blood pressure cuff (SphygmoCor XCEL, AtCor Medical, Sydney, Australia) placed at the brachial artery following established guidelines [35]. Brachial mean arterial pressure (MAP) was calculated as:

1/3 *SBP* + 2/3 *DBP* = MAP.


To estimate central aortic blood pressure, the brachial pressure waveforms were calibrated using cuff-measured brachial systolic and diastolic pressures and then used to generate central aortic pressure waveforms by applying proprietary digital signal processing and transfer function using customized software (SphygmoCor 7.01, AtCor Medical, Sydney, Australia) [35,36], which has been validated in previous work [37,38,39]. Aortic MAP was derived by integrating the area under the aortic blood pressure waveform [12,35].

### 2.5. Heart Rate Variability Recording and Analysis

HRV refers to beat-by-beat variations in time between consecutive heartbeats and is a result of sympathetic and parasympathetic nervous system interactions [24,40]. HRV was continuously recorded for 5 min by electrocardiogram (ECG) using three electrodes placed at the right and left midclavicular lines inferior to the right and left clavicles, respectively, and on the left edge of the rib cage inferior to the pectoral muscles (BIOPAC Systems, Inc., Goleta, CA, USA) [40]. Continuous autonomic data were sampled at 1000 Hz and sent to a data acquisition system (BIOPAC Systems, Inc., Goleta, CA, USA), and data were visually inspected with any artifact removed prior to analysis. HRV was analyzed in the frequency domain (AcqKnowledge 5.0.3 software, BIOPAC Systems, Inc., Goleta, CA, USA) with the following frequency bands: very-low-frequency (VLF, ≤0.04 Hz), low-frequency (LF, 0.04–0.15 Hz), and high-frequency (HF, 0.15–0.4 Hz) [40]. The ratio between low- and high-frequency bands (LF/HF) is used as an index of sympathovagal balance, and the HF band is an indicator of parasympathetic predominance and cardiovagal activity [40]. The VLF and LF bands have been suggested to indicate the vasomotor tone and reflect a combination of parasympathetic and sympathetic activity [41,42,43].

HRV was also analyzed in the time domain (AcqKnowledge 5.0.3 software, BIOPAC Systems, Inc., Goleta, CA, USA) to examine time intervals between successive normal complexes (N-N) [40] using: the standard deviation of the difference between successive N-N (SDSD) and the square root of the mean squared differences of successive N-N intervals (RMSSD). Both the SDSD and RMSSD time domain analysis methods provide estimates of overall HRV without discerning parasympathetic and sympathetic activity or differences in relative contributions [40].

All data acquisition and post-acquisition analyses were carried out in accordance with the Task Force of the European Society of Cardiology and the North American Society of Pacing and Electrophysiology [40]. Logarithmic transformations were performed to induce normality due to the nonnormal distribution of absolute values of HRV parameters [44,45]; therefore, HRV frequency domain measures are presented as normalized units (n.u.).

### 2.6. Baroreflex Sensitivity Recording and Analysis

BRS can be described as alterations of the baroreceptor-heart rate reflex and is the main short-term modulator of arterial blood pressure [23]. Beat-to-beat continuous finger blood pressures were measured from the left middle and index fingers using digital pulse photoplethysmography and calibrated with upper arm blood pressure readings using an internal calibration system (CNAP Monitor 500, CNSystems, Graz, Austria) [46]. Beat-to-beat blood pressures and ECG were simultaneously obtained and recorded by the data acquisition system (BIOPAC Systems, Inc., Goleta, CA, USA).

Five minutes of beat-to-beat series of SBP and R-R intervals were analyzed using the sequence method to obtain spontaneous BRS (AcqKnowledge 5.0.3 software, BIOPAC Systems, Inc., Goleta, CA, USA) [47]. R-R intervals were regressed over SBP for each sequence of 3 or more consecutive cardiac cycles in which both SBP and heart rate increased (up sequences) or decreased (down sequences) in unison using a minimum threshold of 3 milliseconds and 1 mmHg, respectively. The slope of the regression line that relates changes in SBP to changes in R-R intervals was computed, and the total regression for combined up and down sequences was averaged to obtain average BRS [23,47]. Sequences with a minimum r = 0.8 were used. There was a delay in securing the equipment for the HRV and BRS analyses. Therefore, autonomic function variables were only obtained in a subset of participants.

### 2.7. Exercise Protocol

The SRT is a ramp supramaximal test performed on the cycle ergometer and requires participants to maintain 50 revolutions per minute (rpm) while 0.5 kg of weight (25 Watts) is added every 10 s [32]. To minimize any potential learning effects or familiarization with the SRT, a thorough explanation of the test protocol and participants performed a brief warm-up before the test. Following a 2-min warm-up with 0.5 kg of resistance, participants were asked to provide a maximal effort, and the test was terminated when the cadence fell below 50 rpm. Efforts were considered to be maximal when participants showed subjective signs of intense effort (e.g., inconsistent cadence, sweating, facial flushing, and clear unwillingness to continue despite encouragement) [30]. The time until exhaustion and the maximal workload reached were recorded as measures of individual maximal anaerobic capacity.

### 2.8. Statistical Analyses

Descriptive statistics were summarized using mean ± standard errors. Descriptive and baseline variables were analyzed with independent *t*-tests for possible racial differences. A 2-way mixed ANOVA with repeated measures was used to evaluate blood pressure and autonomic recovery responses to acute exercise to test for race and time effects and their interactions (2 × 5, race × time for blood pressure; 2 × 3, race × time for BRS and for HRV). Fisher’s least significant difference (LSD) posthoc analysis was performed when significant factor effects or their interactions were detected. Tests of normality were performed for all variables to assess for normal distribution. HRV data were transformed using logarithmic transformation to achieve normal distribution. For statistical significance, *p* < 0.05 was used, and statistical analyses were performed using IBM SPSS Statistics 26 software (SPSS Inc. Chicago, IL, USA).

### 2.9. Power Analyses

Power analyses were performed for hemodynamic and autonomic variables. Previous data have demonstrated an average difference of 3.75 mmHg with a standard deviation of 1 for aortic blood pressure between Black and White women following maximal aerobic exercise [12]. An estimated sample size of 27 Black and 27 White women would be required to be able to reject the null hypothesis that the population means of the two racial groups are equal with probability (power) 0.8 with a significance level of 0.05. For autonomic variables, an average difference of 9 beats/min in BRS responses to hypertensive stimuli with a standard deviation of 2 between Black and White subjects was observed [28]. An estimated sample size of 22 Black and 22 White women would be required to reject the null hypothesis. For this power calculation, a probability of 0.8 and a significance level of 0.05 were used. Power calculations were performed using PS-Power and Sample Size Program software.

## 3. Results

### 3.1. Participant Characteristics and Exercise Performance

Participant characteristics are presented in Table 1. Twenty (Black: *n* = 8, White: *n* = 12) participants were included in the blood pressure analyses. A subgroup of 10 participants (Black: *n* = 4, White: *n* = 6) was included for autonomic data analyses due to the delay in securing research equipment. There were no significant baseline differences between the two racial groups in age, height, weight, BMI, and MVPA volume. There were no significant differences in resting brachial and aortic SBP, DBP, and MAP or resting frequency domain measures of HRV and BRS between Blacks and Whites (Figure 2A–F and Figure 3A–C, respectively). For each exercise bout, Black and White participants had similar time until exhaustion (Black: 99 ± 4 s and 100 ± 4 s; White: 107 ± 4 s and 107 ± 5 s, for the first and second bout of SRT, respectively) and achieved similar maximal power outputs (Black: 278 ± 10 Watts and 284 ± 9 Watts; White: 290 ± 10 Watts and 294 ± 13 Watts for the first and second bout of SRT, respectively), demonstrating comparable maximal anaerobic capacity between groups. In addition, there were no significant differences in exercise performance between the first and the second bout of SRT in all participants, suggesting the high reliability of this anaerobic performance test.

### 3.2. Racial Differences in BP

Blood pressure values at rest and following exercise are presented in Figure 2A–F. There was no significant race-by-time interaction for the blood pressure variables. There was a significant main effect of time for brachial SBP for all participants (time effect *p* = 0.024, F = 2.998, η^2^ = 0.143, Figure 2A). Brachial SBP was lower at SRT1 P15 (113.9 ± 1.5 mmHg) compared to that at SRT2 P15 (118.7 ± 2.2 mmHg), regardless of race (*p* = 0.033). Brachial SBP was also lower at SRT1 P30 (113.2 ± 1.1 mmHg) compared to that at SRT2 P15 (118.7 ± 2.2 mmHg), regardless of race (*p* = 0.009). There was no main effect of race for brachial SBP. There was a significant main effect of race for brachial MAP (Black: 85.6 ± 1.5 mmHg and White: 81.4 ± 1.2 mmHg, race effect *p* = 0.043, F = 4.767, η^2^ = 0.209, Figure 2C) and brachial DBP (Black: 69.9 ± 1.6 mmHg and White: 65.6 ± 1.3 mmHg, race effect *p* = 0.049, F = 4.476, η^2^ = 0.199, Figure 2B) showing Blacks had 5.06% and 6.57% greater brachial MAP and DBP, respectively, compared to their White counterparts across time. There was a significant main effect of time for brachial DBP (time effect *p* = 0.012, F = 3.468, η^2^ = 0.162, Figure 2B). Brachial DBP was lower at SRT1 P15 (65.9 ± 1.2 mmHg) compared to at rest (69.6 ± 1.2 mmHg), regardless of race (*p* = 0.015). Brachial DBP was also lower at SRT1 P15 (65.9 ± 1.2 mmHg) compared to that at SRT1 P30 (68.5 ± 1.2 mmHg), regardless of race (*p* = 0.012). Brachial DBP was also lower at SRT1 P15 (65.9 ± 1.2 mmHg) compared to that at SRT2 P30 (68.2 ± 1.2 mmHg), regardless of race (*p* = 0.017). There was no time-by-race interaction, main effect of time, or main effect of race for aortic systolic, diastolic, and mean arterial pressures (Figure 2D–F).

### 3.3. Racial Differences in Autonomic Recovery

Autonomic variables at rest and following exercise are presented in Table 2 and Figure 3A–C. There was a significant main effect of time for VLF (F = 6.937, η^2^ = 0.498, Table 2), LF (*p* = 0.009, F = 6.766, η^2^ = 0.491, Table 2), HF (F = 17.351, η^2^ = 0.713, Table 2), and LF/HF (F = 7.174, η^2^ = 0.506, Table 2) for all participants (*n* = 10). VLF (*p* = 0.012), LF (*p* = 0.013), and HF (*p* = 0.006) were reduced at SRT1 P20 compared to that at rest, regardless of race. VLF (*p* = 0.035), LF (*p* = 0.033), and HF (*p* = 0.002) were also reduced at SRT2 P20 compared to that at rest, regardless of race. LF/HF was increased at SRT2 P20 compared to at rest, regardless of race (*p* = 0.023). LF/HF was also increased at SRT2 P20 compared to that at SRT1 P20, regardless of race (*p* = 0.047). There was a significant main effect of time for RMSSD (F = 23.983, η^2^ = 0.774, Table 2) for all participants (*n* = 9). RMSSD was reduced at SRT1 P20 (*p* = 0.001) and at SRT2 P20 (*p* = 0.000) compared to that at rest, regardless of race. There was a significant main effect of time for BRS (F = 8.436, η^2^ = 0.547, Table 2) showing BRS was significantly reduced at SRT1 P20 (*p* = 0.015) and at SRT2 P20 (*p* = 0.006) compared to at rest, regardless of race. There was a significant main effect of time on heart rate (Table 2). Heart rate was significantly increased at SRT1 P20 (*p* < 0.001) and at SRT2 P20 (*p* < 0.001) compared to at rest, regardless of race. Heart rate was also significantly increased at SRT2 P20 compared to SRT1 P20 (*p* = 0.007) and at rest, regardless of race.

We have also explored the potential racial differences in this subset of participants (Black: *n* = 4, White: *n* = 6). There was a significant main effect of race for VLF (Black: 3.12 ± 0.31 n.u. and White: 4.02 ± 0.22 n.u., race effect, *p* = 0.045, F = 5.914, η^2^ = 0.458, Figure 3A) and LF (Black: 3.48 ± 0.24 n.u. and White: 4.64 ± 0.17 n.u., race effect, *p* = 0.006, F = 15.120, η^2^ = 0.684, Figure 3B), showing VLF (*p* = 0.045) and LF (*p* = 0.006) were 22.5% and 24.9% lower, respectively, for Blacks compared to Whites across time. There was also a significant main effect of race for the time domain variables showing RMSSD (Black: 25.4 ± 5.5 ms and White: 46.0 ± 3.9 ms, race effect, *p* = 0.018, F = 6.376, η^2^ = 0.477) was lower for Blacks compared to Whites across time. There was no significant race-by-time interaction for the autonomic variables.

## 4. Discussion

This is, to our knowledge, the first study to compare blood pressure responses following repeated bouts of supramaximal exercise between Black and White women. The main finding of this study revealed racial differences in brachial blood pressure during recovery from supramaximal exercise despite similar resting blood pressure values between Black and White women. This is also the first preliminary study to compare differences in autonomic recovery following a steep ramp anaerobic test between Blacks and Whites. Despite having similar resting autonomic modulation, Blacks had lower VLF and LF bands of HRV during recovery compared to Whites, which may suggest that Blacks exhibit a favorable response following supramaximal exercise. However, given the relatively small sample size and the fact that our study is underpowered, caution should be exercised in generalizing these findings. Future research with a larger sample size will be necessary to confirm and extend these findings.

### 4.1. Racial Differences in Brachial MAP and DBP

The findings from the current study suggest that young, healthy Black women without hypertension have higher brachial MAP and DBP compared to their White counterparts during recovery from repeated bouts of supramaximal exercise despite similar resting blood pressures. This finding is consistent with racial differences that have been reported in existing literature comparing recovery brachial blood pressure responses in Blacks and Whites following aerobic exercise [12,48]. Compared to their White counterparts, Black women tend to exhibit unchanged [48] or exaggerated hemodynamic responses to physiological stressors such as submaximal aerobic exercise [49] and behavioral sympathoexcitation [11]. In agreement with the current study, similar findings of heightened brachial DBP in Blacks were observed following 45 min of acute aerobic exercise performed at 70% HRR at 30, 60, and 90-min post-exercise compared to their White counterparts [12]. Reduced β-adrenergic sensitivity and augmented α1-adrenergic receptor sensitivity in Blacks may be responsible for the exaggerated cardiovascular hyperreactivity [50], resulting in higher vascular resistance and heightened DBP [51]. In addition, limited previous research suggested that the attenuated vasodilation observed in Blacks compared to Whites may be contributed to by lower Nitric Oxide bioavailability [52]. Lower NO bioavailability in the endothelial cells of Blacks compared with Whites may be primarily due to increased O^2−^ production mediated by upregulated NAD(P)H-oxidase activity followed by eNOS uncoupling [53]. Taken together, these differences may lead to heightened total peripheral resistance in Black women following exercise. Cardiac output may also influence blood pressure. However, the observed racial differences in post-exercise blood pressure are not likely due to differences in cardiac output, as cardiac output was not contributing to differential recovery blood pressure between Blacks and Whites following acute aerobic exercise [16]. Our findings may be consistent with a higher risk of CVD in Blacks and partially explained by higher vascular resistance in this racial group [11,49].

Aortic blood pressure has been shown to be more predictive of adverse cardiovascular events and hypertension compared to brachial blood pressure [54]. Aortic blood pressure was similar between Black and White women at rest and following exercise in the current study, which is in contrast to the findings of previous investigations [12,48,55]. Racial differences in aortic blood pressure have been observed at rest in Black and White men [55] and following acute maximal aerobic exercise, with Black men and women showing higher [12] or unchanged [48] aortic blood pressure during recovery compared to their White counterparts. The discrepancies in aortic blood pressure response between the current study and previous work may be explained by cardioprotective female sex hormones, which have been shown to increase vascular smooth muscle responsiveness and have blood pressure-lowering effects in women [56]. These discrepancies may also be due to differences in exercise volume, which has shown to be important in eliciting transient post-exercise blood pressure reductions [57], thus repeated steep ramp tests in the current study may not have been a sufficient stimulus to elicit changes in aortic blood pressure in young, healthy Black and White women. The prolonged hypotensive effect of regular exercise training may be due to repeated instances of post-exercise blood pressure reduction, with preliminary data suggesting the magnitude of the acute blood pressure lowering with exercise may predict the extent of blood pressure lowering after chronic training interventions in prehypertensive Whites [14]. If such a correlation also exists in Blacks following anaerobic exercise, those that exhibit greater post-exercise hypotension may benefit the most from this type of exercise prescription. Since the two bouts of steep ramp supramaximal test utilized in the current investigation did not produce post-exercise blood pressure reduction in Black women, additional volume or alternative excise type might be explored if antihypertension is the primary goal of the exercise training program. Therefore, understanding the dose-response relationship between supramaximal exercise volume and blood pressure changes may be important for outlining high-intensity interval exercise recommendations and prescribing adequate exercises doses to elicit potential desired health benefits.

Further research is warranted on the effects of different anaerobic exercise volumes on both brachial and aortic blood pressures during recovery in Blacks and Whites, which bears clinical relevance in populations that exhibit disparities in CVD as well as translational applications with the increasing popularity of interval exercise programs. Despite the inclusion of vigorous-intensity exercise in national physical activity guidelines and recommendations [18], these focus on aerobic exercise and fail to capture anaerobic modalities. Therefore, future research examining the effects of anaerobic exercise on blood pressure would be important for further developing physical activity guidelines for healthy populations and groups that exhibit different CVD risks.

### 4.2. Post-Exercise Autonomic Recovery and Racial Differences in HRV

All participants exhibited significant decreases in all frequency domain and time domain measures of HRV, which reflect the relative contributions of the autonomic nervous system branches, following repeated bouts of steep ramp supramaximal test compared to baseline. The decrease in time domain measures at 20 min following repeated bouts of supramaximal exercise was in agreement with previous research on the recovery timeline following acute supramaximal exercise [58]. Sympathovagal balance, indicated by LF/HF, was significantly increased following repeated exercise bouts compared to baseline and is consistent with previous studies that have shown temporal alterations in HRV and sympathovagal balance following high-intensity aerobic and supramaximal exercise [19,41,44]. This supports the idea that autonomic modulation is altered through reduced vagal reactivation and increased sympathovagal balance following supramaximal exercise in young, healthy women without hypertension. Frequency domain analysis of HRV offers a non-invasive assessment of autonomic modulation in healthy individuals that exhibit different CVD risks as well as clinical populations that may be at risk for adverse cardiovascular events [40].

This is the first study to compare differences in autonomic recovery following exercise in Blacks and Whites. It was hypothesized that Black women would exhibit lower sympathovagal balance and impaired BRS and HRV responses following repeated bouts of acute supramaximal exercise compared to White women. Racial differences were observed in HRV time domain measures and some HRV frequency domain measures, as Blacks exhibited lower power in the VLF and LF bands, which reflect a combination of sympathetic and parasympathetic activity [40] compared to their White counterparts. The VLF band has been suggested to be associated with the peripheral vasomotor tone, which plays a role in blood pressure reductions and decreases in vasomotor tone may be associated with concomitant reductions in MAP [42,43]. In contrast to the initial hypothesis, the reduced VLF power in Blacks suggests this group may exhibit favorable changes in autonomic modulation following acute supramaximal exercise and may benefit from engaging in maximal or high-intensity interval exercise. Low VLF power has been associated with increased chronic inflammation [59], and inflammation may contribute to the observed racial differences in VLF. One potential mechanism for higher VLF power in Whites may be due to transiently elevated inflammatory markers (e.g., IL-6) following exercise. The renin-angiotensin system (RAS) has been shown to be related to the VLF. The RAS responds to exercise stress stimulation and increases inflammation to activate the sympathetic nerves [60]. While no study has examined the potential racial differences in inflammatory markers in Blacks and Whites following acute exercise, Heffernan et al. reported decreased inflammatory markers and increased measures of vagal modulation following resistance training in Black men [61]. On the other hand, the White men did not experience improvements in vagal modulation or decreases in inflammatory markers with resistance training, possibly due to their low baseline levels of inflammatory markers [61]. Therefore, supramaximal excise may offer greater health benefits in Blacks compared to Whites and should be considered when performing personalized exercise prescriptions.

Differences in autonomic modulation cannot be attributed to resting differences, as all baseline HRV and BRS values were similar between Blacks and Whites. It is worth noting that no racial differences were observed in HF power during recovery, suggesting similar levels of parasympathetic modulation between racial groups. This is a novel observation, as existing evidence suggests that young Blacks tend to exhibit lower parasympathetic activity and decreased HRV at rest [25,26,27]. Recovery HRV has been utilized as a biomarker in previous investigations to assess, monitor, and adjust training loads in athletes to optimize training and balance recovery timing using an individualized approach [62,63]. Therefore, future research is warranted to explore potential racial differences in autonomic recovery from supramaximal exercise, which may assist in evaluating the risks and benefits of this type of exercise and in developing individualized exercise training prescriptions.

BRS reflects alterations of the baroreceptor-heart rate reflex and is mediated by vagal activity. In this study, BRS was significantly reduced following single and repeated bouts of acute supramaximal exercise in all participants, which is in agreement with previous studies that have shown BRS is reduced following supramaximal anaerobic exercise, high-intensity interval training and maximal aerobic exercise [19,41,57]. These findings may be explained by suppressed vagal reactivation following supramaximal exercise and the resetting of BRS to a higher operational point during exercise in both Black and White women [64]. Despite emerging evidence of impaired BRS in young, healthy normotensive Blacks compared to their White counterparts at rest [22,26,27,28] and following hypotensive and hypertensive stimuli [28], there were no racial differences in BRS in the current study, which may be explained by the small sample size for autonomic modulation data.

### 4.3. Clinical Implications

High-intensity interval exercise has been demonstrated as an effective training method [65]. However, the greater hemodynamic strain posed by high-intensity exercise remains contentious. The current study shows that repeated supramaximal exercise elicited heightened brachial blood pressure and lower HRV during recovery in young and healthy Black women compared to their White counterparts. Impaired BP recovery after acute exercise in midlife may be a marker of subclinical and clinical CVD and mortality in later life [66]. This may suggest a need to adapt repeated supramaximal exercise protocol before their application in diseased populations. Some possible adaptations may include longer rest periods between exercise intervals and reduced intensity levels to limit potentially negative effects.

### 4.4. Limitations 

We acknowledge several potential limitations of the present study. First, the current autonomic data should be interpreted with caution due to the small sample size; however, our findings add new information to the autonomic modulation literature regarding potential racial differences during recovery from exercise. Future studies with larger sample sizes are warranted to support these findings. Second, while the respiratory rate may influence BRS and HRV, it was not controlled for in this study as paced breathing has been shown to reduce blood pressure and stimulate the parasympathetic nervous system [67]. Third, while direct arterial catheterization is considered the gold standard for aortic blood pressure assessment, a generalized transfer function using pulsations taken at the brachial artery was used to derive aortic blood pressure in the current study. However, the non-invasive generalized transfer function method has been validated in measuring aortic blood pressure [37,38]. In addition, we did not control for phases of the menstrual cycle in our study. Despite the potential impact of female hormone fluctuations on hemodynamic variables, the influence of fluctuations in sex hormones across the menstrual cycle on overall vascular function remains inconclusive [68], and there is reported evidence of the minimal impact of the menstrual cycle on post-exercise hemodynamics [69]. Further study is warranted to investigate the impact of the menstrual cycle on potential racial differences in blood pressure and autonomic recovery. Furthermore, we did not include a comprehensive evaluation of psychological stress, given the scope of our study. Chronic psychological stress disproportionately affects Black women and may play a role in the racial differences we observed. Lastly, only young, healthy women without hypertension were included in this study; therefore, caution is warranted in extrapolating these findings to men or individuals with hypertension.

## 5. Conclusions

Compared to their White counterparts, young, healthy Black women without hypertension exhibited heightened brachial blood pressure during recovery from repeated bouts of acute supramaximal exercise, which may be associated with an increased cardiovascular risk. In addition, preliminary findings from this study with a subset of participants indicate that Black women may exhibit decreased HRV during recovery from supramaximal exercise compared to Whites. This suggests that Blacks may benefit from engaging in this type of exercise; however, additional research with larger sample sizes is needed to support these findings. The mechanisms explaining these racial differences in autonomic recovery remain elusive. Future research may provide additional knowledge regarding blood pressure and autonomic responsiveness, or lack thereof, to different exercise intensities and may assist in creating more effective and tailored exercise prescriptions and outlining preliminary recommendations for supramaximal exercise for inclusion in national physical activity guidelines.

## Figures and Tables

**Figure 1 ijerph-20-05615-f001:**
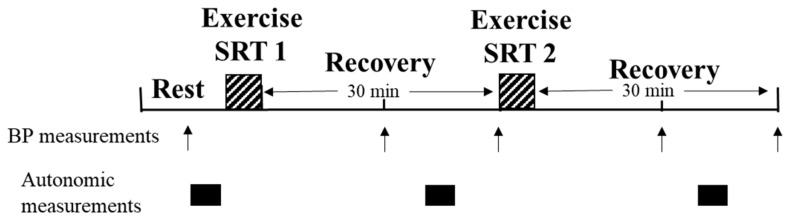
Schematic of testing and measurement protocol. SRT, steep ramp test; SRT1, first steep ramp test; SRT2, second steep ramp test; BP, blood pressure.

**Figure 2 ijerph-20-05615-f002:**
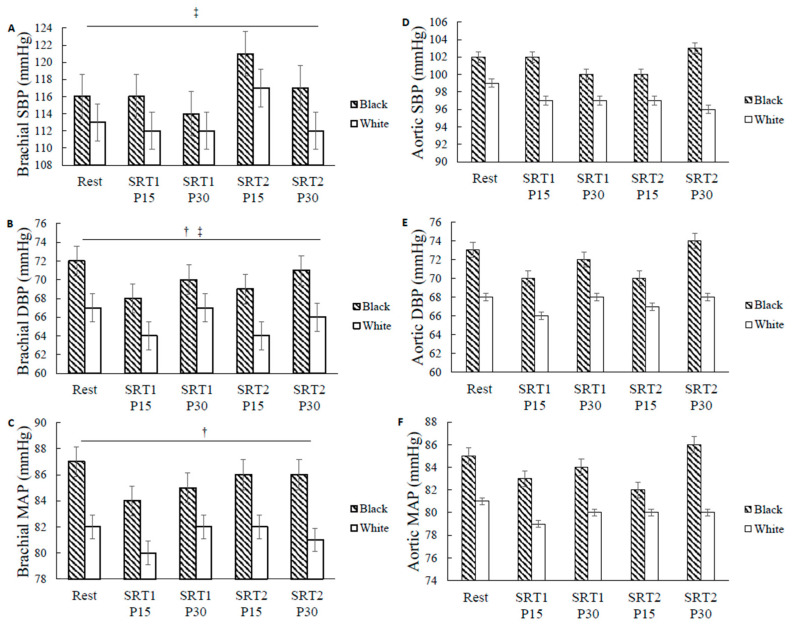
Brachial and aortic blood pressure at rest and during recovery in Black and White women. Brachial (**A**) systolic blood pressure (SBP), (**B**) diastolic blood pressure (DBP), (**C**) mean arterial pressure (MAP) and aortic (**D**) SBP, (**E**) DBP, and (**F**) MAP quantified in millimeters of mercury (mmHg) at rest and at 15 and 30 min after repeated bouts of steep ramp supramaximal test (SRT1 and SRT2). Values are means ± SE. † *p* < 0.05, significant race differences. ‡ *p* < 0.05, significant effect of time.

**Figure 3 ijerph-20-05615-f003:**
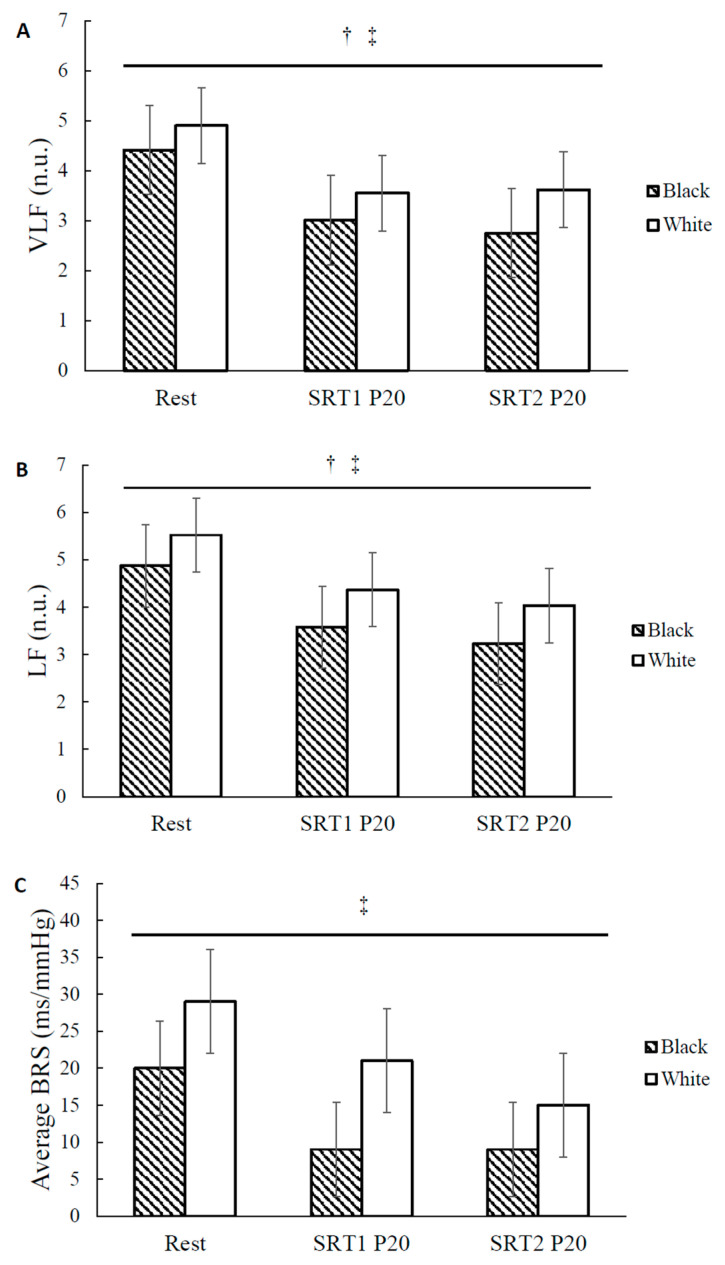
Autonomic modulation at rest and during recovery in Black and White women. (**A**) Very-low-frequency (VLF) and (**B**) low-frequency (LF) power of heart rate variability (HRV)quantified in normalized units (n.u.) and (**C**) average baroreflex sensitivity (BRS) quantified in milliseconds of RR interval prolongation per change in arterial pressure (ms/mmHg) at rest and at 20 min after repeated bouts of steep ramp supramaximal test (SRT1 and SRT2) in Black and White women. Values are means ± SE. † *p* < 0.05, significant race differences. ‡ *p* < 0.05, significant effect of time.

**Table 1 ijerph-20-05615-t001:** Participant Characteristics.

	Black (*n* = 8)	White (*n* = 12)	*p*-Value
Age (years)	22 ± 1	22 ± 1	0.82
Height (cm)	167.3 ± 2.2	165.5 ± 1.5	0.48
Weight (kg)	69.2 ± 3.8	63.0 ± 3.2	0.23
BMI (kg/m^2^)	24.7 ± 0.7	22.9 ± 0.8	0.83
MVPA volume (min/wk.)	135.0 ± 43.3	179.6 ± 47.3	0.37

Values are mean ± SE; *n*, No. of participants; cm, centimeters; kg, kilograms; BMI, body mass index; kg/m^2^, kilograms divided by height in meters squared; MVPA, moderate-vigorous physical activity; min/week, minutes per week.

**Table 2 ijerph-20-05615-t002:** Autonomic Modulation at Rest and During Recovery for All Participants (*n* = 10 for BRS and frequency-domain variables, and *n* = 9 for time-domain variables).

	Rest	SRT 1 P20	SRT 2 P20	*p*-Value
BRS (ms/mmHg) ^‡^	26.3 ± 7	16.7 ± 4	13.1 ± 3	0.004
VLF (n.u.) ^‡^	4.35 ± 0.23	3.18 ± 0.26	3.18 ± 0.34	0.008
LF (n.u.) ^‡^	4.91 ± 0.28	3.65 ± 0.23	3.63 ± 0.30	0.009
HF (n.u.) ^‡^	5.6 ± 0.2	3.5 ± 0.4	2.7 ± 0.5	0.000
LF/HF ^‡^	0.8 ± 0.2	1.8 ± 0.3	3.8 ± 1.0	0.007
RMSSD (ms) ^‡^	59.5 ± 2.7	26.5 ± 6.0	21.2 ± 4.8	0.000
Average HR (bpm) ^‡^	63 ± 4	75 ± 5	79 ± 6	<0.001

Values are mean ± SE; *n*, No. of participants; SRT1, first bout of steep ramp test; SRT2, second bout of steep ramp test; P20, 20 min post-exercise; BRS, baroreflex sensitivity; VLF, log transformation of the very-low-frequency band; LF, log transformation of the low-frequency band; HF, log transformation of the high-frequency band; LF/HF, low-to-high frequency ratio; RMSSD, the square root of the mean squared differences of successive N-N intervals; Average HR, average heart rate. ^‡^ *p* < 0.05, significant effect of time.

## Data Availability

Data are available upon request due to privacy/ethical restrictions.

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
