# Peer review of "Racial Differences in Blood Pressure and Autonomic Recovery Following Acute Supramaximal Exercise in Women"

_ijerph, 2023, doi:10.3390/ijerph20095615_

Round 1
Reviewer 1 Report (New Reviewer)
This is an interesting study and has a novelty of blood pressure at rest and automatic system response during high-intensity exercise in black and white women.
However, there are several problems to solve.
Major comments
There are too small participants to demonstrate the different racing BP and autonomic nervous system after moderate- and vigorous intensity exercise program.
Minor comments
1) In Figure 1, the authors should attach the abbreviation of SRT in foot.
2)In Table 1, the authors should add the P value which compare between Black and White women.
3)In Table2, number of patients might describe mistake as there are comments at right out-side. Is it correct?
4) Why don’t the authors compare the effect between black and white women only in Table 2? The reader might not understand easily the meaning as above the reason in the current study.
5) In figure 2A, brachial SBP of SRT2P15 is only high. The author had better describe the sentence in the text.
Author Response
Please see the attachment.

Reviewer 2 Report (New Reviewer)
This study is a well written and conducted study. Great care was included within the literature background and study methodology. However, there are deficiencies that exist that have the potential to be significant, I believe can be mitigated by the authors being more transparent in their presentation of the methods. For instance, the sample size is small (with a sub sample that is even smaller to address the autonomic influence) without a power analysis. Despite this, the authors were able to achieve significance (p < 0.05). This statistical achievement is representative of the methods, but the F values and effect sizes are missing from the article which will provide a better indicator of the author’s thesis, which is the effect of race on post exercise hemodynamic recovery. Additionally, while the methodology regarding the hemodynamic testing is vast, none of the citations are within 10yrs. I have left more specific comments below:
ABSTRACT-
P1ln15- please include description/methodology of mode of exercise, autonomic modulation and blood pressure
P1ln19- when was MAP and brachial pressures high, at rest, during exercise, after exercise?
INTRODUCTION
P1ln44- this statement is not true. I searched the phrase, “maximal anaerobic exercise on cardiovascular hemodynamics” and 1.3M articles are relevant to this topic.
METHODS
P2ln55- what type of exercise?
P2ln97- was a power analysis performed to ensure objective evaluation of the research problem? There seem to be many variables accounted for while the sample size is small and unbalanced (black n=8; white = 12), especially considering the autonomic measurements which were only conducted in a subset of participants (black n=4; white n=6). Please include the power analysis to show the research design is adequate to evaluate the research problem.
P3ln98- there is much discussion regarding the effect of stress and impaired hemodynamics within blacks, particularly black women. However, there is no attempt by the authors to explore current psychological stress among their population sample which significantly impairs internal validity. While the authors seemed to have taken great care in ensuring the homogeny of their cross-sectional sample, this is a glaring vacancy among the methodology, especially considering the sensitivity of HRV, vascular function, etc to psychological stress.
Pg3ln135-139- the research cited is all older than 10yrs. While some of the citations are immediately relevant to the technology, more recent publications outline methods using the nomenclature such as pulse wave velocity. If this is the application the authors are implementing, please revise as necessary. If not, than please include more recent publications to validate the current method to ensure it is relevant and/or still valid according to more recent methodologies.
P5ln190-195- novice cyclists rarely can accomplish maximum due to local fatigue. Was there a familiarization protocol prior to the testing or please include publications that show peak CV values/VO2 values have been achieved in novice cyclists?
RESULTS
Please include F values and effect sizes (eta squared) for all ANOVA models.
There is much redundancy regarding the results section as it is written currently. Opposed to writing the mean values for each DV, please revise the tables and eliminate the verbiage to simplify the presentation of the results.
While the autonomic variables are an enticing addition to the paper, it seems as an after thought as only a subgroup of the sample (which is already small) is included (and only after the 1st exercise bout). Please justify why only a sub-sample was used to evaluate this component of the study, which is a central component of the results/conclusions.
Author Response
Please see the attachment.

Reviewer 3 Report (New Reviewer)
The manuscript contains information that could probably be of significance to the particular field of study. However, the submitted file-manuscript contains comments probably made during draft preparation. Thus I cannot proceed with a review at this stage. I advise that the authors carefully correct this and resubmit their manuscript for review.
Author Response
Please see the attachment.

Reviewer 4 Report (New Reviewer)
The manuscript is generally well written and presented, but there are several major concerns, including the methodology used for the study.
1. The study design is creative to use the SRT, a nice standardized protocol. The authors consider this test a “maximal anaerobic test”. Note: this test is being used as a test for maximal aerobic capacity in both pediatric and adult to get around requiring apparatus for measuring oxygen consumption. It has a large anaerobic component, but it isn’t a pure anaerobic test. Suggest revising.
2. The timeline while accurate is misleading physiologically. Peak exercise data are not provided (and represent large increments over rest. HR ~87% of max.) See normative data for extent of the physiological perturbation on HR and BP. https://www.ncbi.nlm.nih.gov/pmc/articles/PMC6969871/pdf/421_2019_Article_4255.pdf
3. The exercise peak data are what provide the degree to which these Black and White women may differ based on the true “starting” point for recovery measurements. Yes, the rest are important too, but the interpretation is difficult without knowing the exercise starting point. Moreover, given the interest in autonomic function, the only relatively “pure” assessment is the initial heart rate recovery (HRR) to index parasympathetic reengagement.
4. Another issue with the design is the way it is implemented. A HIIT protocol would only give these participants < 4 minutes of recovery before the repeat SRT. It would require several. If it was just a standalone SRT plus recovery it would make more sense than the way the protocol is carried out.
5. The HRV on so few of the participants (only 4 of the Black women) is odd, if this was part of the protocol, why aren’t all of the data represented. If there were technical difficulties. It raises the question of whether they were really technical difficulties or whether some other post exercise CV issue is going on. —These data examined by racial differences are too thin.
Based on Table 2 (all participants combined) Parasym. Reengagement appears to be faltering. This has been demonstrated early after repeat sprint and HIIT compared to moderate intensity exercise. It could be a very important health parameter if distinct between Black and White women given their disparate CV risk profiles https://journals.physiology.org/doi/epdf/10.1152/ajpheart.00062.2007.
Specific Issues in presentation
· The graphs should be bar graphs the “time” doesn’t make sense without the peak exercise values
Round 2
Reviewer 2 Report (New Reviewer)
Thank you to the authors for their efforts to address the reviewer comments. However, the misfortune of COVID-19 does not excuse the authors from following both scientific president/best practices. I agree the authors have designed and executed a strong contribution to the science. However, it is also noted the authors have not been completely transparent with their sample and therefore results, which should be included throughout to maintain objectivity. I have left more specific comments below in italics:
Previous comments: “P2ln97- was a power analysis performed to ensure objective evaluation of the research problem? There seem to be many variables accounted for while the sample size is small and unbalanced (black n=8; white = 12), especially considering the autonomic measurements which were only conducted in a subset of participants (black n=4; white n=6). Please include the power analysis to show the research design is adequate to evaluate the research problem. Response: Thanks for raising your concern about the power analyses. We have performed a power analysis based on the primary outcomes. For blood pressure, we determined an estimated sample size of 27 Black and 27 white participants would to be able to reject the null hypothesis. An estimated sample size of 22 Black and 22 white women was calculated for autonomic variables. However, we had a delay in acquiring the equipment for autonomic variables, which prevented us from obtaining data in all participants. In addition, while we planned to recruit and test additional participants, our campus was closed for two years due to COVID-19. We considered resuming participant recruitment after COVID-19 related restrictions were lifted. However, there is emerging research suggesting a transient impact of COVID-19 on autonomic function (Skow RJ et al, 2022 https://pubmed.ncbi.nlm.nih.gov/36331556/) and we are hesitant to combine our participants if we were to continue recruitment. Despite these setbacks, we believe our results still provide valuable insights into the autonomic modulation literature regarding potential racial differences during recovery from exercise and contribute to the existing literature in a meaningful way. And as you noted, we analyzed our data and were able to achieve statistical significance (p < 0.05).”
Please make sure and provide language detailing how your sample was collected within the manuscript. While it is underpowered, you were still able to achieve significance. However, it will be important to communicate that this is a convenience sample rather than one that is adequately powered.
All tables/figures include abbreviations that are not clear unless the reader sources them from the manuscript. Please clarify all abbreviations within the figures/tables.
Previous Comments: “P5ln190-195- novice cyclists rarely can accomplish maximum due to local fatigue. Was there a familiarization protocol prior to the testing or please include publications that show peak CV values/VO2 values have been achieved in novice cyclists? Response: We did not perform a familiarization session for cycle ergometer. However, to minimize any potential learning effects or familiarization with the SRT, we provided a thorough explanation of the test protocol and allowed participants to perform a brief warm-up before the test. In addition, we reported no significant differences in exercise performance between the first and the second bout of SRT in all participants, suggesting high reliability of this anaerobic performance test. Therefore, we feel confident that the lack of familiarization session did not negatively impact our study. Regarding the validity of steep ramp test, we are unable to find any literature specifically in novice cyclists, however, there are strong correlations observed between peak power output from SRT and VO2max from standard VO2peak tests in cancer survivors (e.g. Weemaes PT et al, 2021https://www.sciencedirect.com/science/article/pii/S0003999321003750 and Stuiver MM et al, 2017 https://www.sciencedirect.com/science/article/abs/pii/S0003999317301582). Therefore, it is reasonable to assume that the SRT can also be a valid and reliable test in novice cyclists.”
Thank you for providing reasonable justification for your exercise protocol. However, both the studies you cited as well as the language to justify the protocol is not found within your revised manuscript. Furthermore, while the Stuiver et al. article is shown within the references, it is not found intext. Considering the vast number of references and at least one that is within the references and not intext, please review all references and ensure that each of those references are found within the article. Also, not sure if there is a reference maximum but 92 is exorbitant for a primary research article.
Previous comments: “While the autonomic variables are an enticing addition to the paper, it seems as an after thought as only a subgroup of the sample (which is already small) is included (and only after the 1st exercise bout). Please justify why only a sub-sample was used to evaluate this component of the study, which is a central component of the results/conclusions. Response: Thanks for your enthusiasm towards the autonomic variables. We have provided our justifications in our response above regarding power analysis. Just to clarify the timeline of measurements, since it takes longer to calibrate and obtain autonomic function variables, we conducted only one measurement after each exercise bout. Therefore, we have reported baseline, recovery from the first exercise bout, and recovery from the second exercise bout for autonomic variables.”
The reviewer does acknowledge the author’s justification. However, the autonomic measurements were only included on a subset of the sample while the conclusions assert otherwise. Therefore, please include/strengthen the language regarding how and why the sample underpowered and the subset throughout the methods, results, and conclusions.
Author Response
Please see the attachment.

Reviewer 3 Report (New Reviewer)
The authors have substantially improved the manuscript in terms of methodology and data reporting compared to the earlier submitted version.
The authors state in the limitations section the small sample size of the study. However, no power analysis was conducted. Thus, I suggest that the authors include the power calculations in the statistical analysis section of the manuscript and state whether their study is underpowered or not and discuss possible research implications.
In the discussion, the authors should elaborate on a greater extent regarding the physiological mechanisms that are implicated in the observed effects on blood pressure and autonomic recovery following exercise.
Another issue is the quality of the figures. I suggest revising the graphs and improving the resolution for better readability.
Author Response
Please see the attachment.

Reviewer 4 Report (New Reviewer)
The revision has improved the manuscript. The revision doesn't change the fact that while the HRV data are sound, there is insufficient sample size to split by race.
Suggest revising in a manner to retain the figure as is, but discussing the main effects (n=10) first —Separating out the preliminary data split by race at the end of this part of the results in a simplified fashion, including restating the n= 4 and n=6. The discussion should then be adjusted accordingly.
minor
—Consistent with Black, when stating white the "W" should be capitalized
—Search anaerobic, to ensure it has been changed where intended throughout
Author Response
Please see the attachment.

This manuscript is a resubmission of an earlier submission. The following is a list of the peer review reports and author responses from that submission.
Round 1
Reviewer 1 Report
Bajdek et al. report racial differences in blood pressure and autonomic recovery following acute maximal anaerobic exercise in women. This study is potentially interesting. However, there are several concerns listed below.
This paper will be strengthened by addressing the following issues.
- The authors should give more detailed explanation about relationship between anaerobic exercise and CVD risk.
- Although the authors suggest that black may benefit from engaging in anaerobic exercise, they should give more detailed discussions for racial differences. What is the important physiological meaning for racial differences in response to maximal anaerobic exercise.
Author Response
We would like to thank the editor and the reviewers for their careful evaluation of our manuscript, along with their valuable comments and suggestions, which we believe have greatly improved our manuscript. Below are the specific responses to each of the comments.
Bajdek et al. report racial differences in blood pressure and autonomic recovery following acute maximal anaerobic exercise in women. This study is potentially interesting. However, there are several concerns listed below.
This paper will be strengthened by addressing the following issues.
- The authors should give more detailed explanation about relationship between anaerobic exercise and CVD risk.
Response: Thank you for this suggestion. We have updated the 1st paragraph in the introduction and in the discussion to better tie high-intensity interval exercise with anaerobic exercise and cardiovascular adaptions.
- Although the authors suggest that black may benefit from engaging in anaerobic exercise, they should give more detailed discussions for racial differences. What is the important physiological meaning for racial differences in response to maximal anaerobic exercise.
Response: This is a great suggestion. We have added more discussion about the physiological and practical meaning for racial differences in response to maximal anaerobic exercise in the revised manuscript (line 301-309, 328-335, and 374-385).
Reviewer 2 Report
General comments:
The article investigated Racial Differences in Blood Pressure and Autonomic Recovery Following Acute Maximal Anaerobic Exercise in Women. It is clear that considerable efforts have been invested in this project. But I think there are some improvements that could be made to highlight the novelty factors of this study. The below comments are intended to explain this view and are hoped to be of benefit to the authors.
Specific comments
Introduction:
- The authors reported interesting findings from the literature on the impact of traditional aerobic training programs on cardiovascular hemodynamics or autonomic modulation, but they did not explain sufficiently the underlying mechanisms of these changes, it is necessary to develop this part to clarify the backdrop of what is already known/not known and how these physiological changes can influence our practice?
- It is necessary to provide a clear definition of what the authors consider anaerobic exercise
-L42-48: Authors reported racial differences in blood pressure and autonomic modulation, but they don’t explain the underlying mechanism of these difference
-L52-54: “While this acute blood pressure reduction has been consistently observed in whites following acute aerobic exercise, it appears to be absent in Black individuals” How do the authors explain this finding?
-L64-67: This paragraph describes the parameters that evaluate the autonomic nervous system, I think it should be moved to the material and method part.
-L81-87: It is important to explain the rationale of this study, but the purpose needs to be more concise. How does this analysis advance our knowledge about Racial Differences in Blood Pressure and Autonomic Recovery following anaerobic exercise? And How it can change our practice?
Methods:
L91-99: Fitness level was not mentioned in the selection criteria, blood pressure and autonomic nervous system may also depend on fitness level so this is a very important selection criterion, was it assessed? if yes, please add the data.
L129: What did the author mean by a transfer function, provide please more details about the used function to calculate the central aortic blood pressure; it was used in which context, in the healthy subjects or in patients? With a specific software?
L133-150: For HRV analysis, authors reported only frequency domain analysis, why they did not use time-domain analysis, which could provide interesting parameters such as RR intervals, RMSSD, SDNN, SD1, and SD2. I think that frequency-domain analyses are insufficient and I propose to add time-domain analysis in order to have a global analysis of the autonomic nervous activity
L151: Did the authors control the respiratory rate during the BRS measurement? Did they use spontaneous or controlled breathing to eliminate respiratory baroreflex-non-related influence and resonance effect on heart rate fluctuations?
L171-172: “participants were asked to provide a maximal effort and the test was terminated when cadence fell below 50 rpm” How did the authors assess the maximal effort of the subjects? Did they use the Perceived Exertion scale to assess their maximal effort?
Discussion:
-Please consider including possible mechanisms explaining the racial differences you observed in blood pressure and autonomic nerve recovery, the authors recite the study results in the discussion portion and compare their results with what is observed in previous studies they do not really present a mechanistic discussion.
Reviewer 3 Report
98-99: Participants taking any medications other than combined estrogen and/or progestin hormonal contraceptive therapy were excluded from participation.
It is well known how strongly estrogen and progesterone influence sports performance. In this manuscript, not only were women using hormonal contraception allowed to study, but the groups of participating women were not hormonally balanced (they were not in the same phase of the menstrual cycle). This is a critical methodological error, which makes the research results unreliable, and thus the present manuscript does not meet the fundamental conditions for its publication.
Author Response
We would like to thank the editor and the reviewers for their careful evaluation of our manuscript, along with their valuable comments and suggestions, which we believe have greatly improved our manuscript. Below are the specific responses to each of the comments.
98-99: Participants taking any medications other than combined estrogen and/or progestin hormonal contraceptive therapy were excluded from participation.
It is well known how strongly estrogen and progesterone influence sports performance. In this manuscript, not only were women using hormonal contraception allowed to study, but the groups of participating women were not hormonally balanced (they were not in the same phase of the menstrual cycle). This is a critical methodological error, which makes the research results unreliable, and thus the present manuscript does not meet the fundamental conditions for its publication.
Response: We appreciate the comment and agree that cyclic changes in estrogen and progesterone concentration may influence some of the vascular measurements. We fully acknowledge the limitation of our study and updated our limitation section. In our upcoming studies, female participants will be tested during early follicular phases (self-report, days 1–5) of their menstrual cycle or placebo phases of the oral contraceptive cycle to control for the influence of hormone fluctuations.
Despite this limitation, we would like to respectfully point out that previous studies evaluating the influence of fluctuations in sex hormones across the menstrual cycle on overall vascular function remain inconclusive [5,6]. In addition, a recent study by D’Agata et al. demonstrate a significant effect of race on peripheral microvascular function such that Black women exhibit significant attenuations in microvascular function across the menstrual cycle compared with white women [7], suggesting the racial disparity might be a more potent modulator than female hormone fluctuations. Therefore, our exploratory project warrants future investigations on the impact of menstrual cycle on potential racial differences in blood pressure and autonomic recovery.
- Lynn, B.M.; McCord, J.L.; Halliwill, J.R. Effects of the menstrual cycle and sex on postexercise hemodynamics. Am J Physiol Regul Integr Comp Physiol 2007, 292, R1260-1270, doi:10.1152/ajpregu.00589.2006.
- Williams, J.S.; Dunford, E.C.; MacDonald, M.J. Impact of the menstrual cycle on peripheral vascular function in premenopausal women: systematic review and meta-analysis. American journal of physiology. Heart and circulatory physiology 2020, 319, H1327-H1337, doi:10.1152/ajpheart.00341.2020.
- D'Agata, M.N.; Hoopes, E.K.; Berube, F.R.; Hirt, A.E.; Kuczmarski, A.V.; Ranadive, S.M.; Wenner, M.M.; Witman, M.A. Evidence of reduced peripheral microvascular function in young Black women across the menstrual cycle. Journal of applied physiology 2021, 131, 1783-1791, doi:10.1152/japplphysiol.00452.2021.
Round 2
Reviewer 2 Report
The authors resubmitted their study focus on “Racial differences in blood pressure and autonomic recovery following acute maximal anaerobic exercise in women”. Overall, the authors have taken into consideration all the suggestions and have massively enhanced the quality of the manuscript, but the manuscript still needs improvements to be publish in IJERPH. The below comments are intended to explain this view and are hoped to be of benefit to the authors.
Results:
Authors should add the time-domain analysis in order to have a global analysis of the autonomic nervous activity.
Discussion:
The mechanistic discussion need to be improved.
The authors did not elaborate the clinical applicability of their findings.